# Modeling of Organic Fouling in an Ultrafiltration Cell Using Different Three-Dimensional Printed Turbulence Promoters

**DOI:** 10.3390/membranes13030262

**Published:** 2023-02-23

**Authors:** Szabolcs Kertész, Nikolett Sz. Gulyás, Aws N. Al-Tayawi, Gabriella Huszár, József Richárd Lennert, József Csanádi, Sándor Beszédes, Cecilia Hodúr, Tamás Szabó, Zsuzsanna László

**Affiliations:** 1Department of Biosystems Engineering, Faculty of Engineering, University of Szeged, Moszkvai krt. 9, H-6725 Szeged, Hungary; 2Doctoral School of Environmental Sciences, University of Szeged, Tisza Lajos krt. 103, H-6725 Szeged, Hungary; 3Department of Physical Chemistry and Materials Science, University of Szeged, Rerrich Béla tér. 1, H-6720 Szeged, Hungary; 4Faculty of Automotive Engineering, Széchenyi István University, Egyetem tér. 1, H-9026 Győr, Hungary; 5Department of Food Engineering, Faculty of Engineering, University of Szeged, Moszkvai krt. 9, H-6725 Szeged, Hungary

**Keywords:** 3DP turbulence promoters, membrane fouling, resistance-in-series model, Hermia models, ultrafiltration

## Abstract

Designing turbulence promoters with optimal geometry and using them for ultrafiltration systems has been a key challenge in mitigating membrane fouling. In this study, six different turbulence promoters were created using three-dimensional printing technology and applied in dead-end ultrafiltration. Three-dimensional-printed (3DP) turbulence promoter configurations were integrated into a classical batch ultrafiltration cell. The effects of these configurations and the stirring speeds on the permeate filtration flux, organic rejections, and membrane resistances were investigated. The fouling control efficiency of the 3DP promoters was evaluated using two polyethersulfone membranes in a stirred ultrafiltration cell with model dairy wastewater. The Hermia and resistance-in-series models were studied to further investigate the membrane fouling mechanism. Of the Hermia models, the cake layer model best described the fouling in this membrane filtration system. It can be concluded that the 3DP turbulence promoters, combined with intense mechanical stirring, show great promise in terms of permeate flux enhancement and membrane fouling mitigation. Using a well-designed 3DP turbulence promoter improves the hydrodynamic flow conditions on the surface of the stirred membrane separation cells based on computational fluid dynamics modeling. Therefore, the factors effecting the fabrication of 3DP turbulence promoters are important, and further research should be devoted to revealing them.

## 1. Introduction

The dairy industry generates a huge volume of wastewater originating from washing and cleaning operations and its large water consumption [1]. This wastewater contains high concentrations of organic matter, suspended particles, fat, oil, and grease, which has to be treated effectively [2]. Membrane filtration processes, such as ultrafiltration (UF) and nanofiltration, are promising methods for treating industrial dairy wastewater [3]. They have several advantages compared to other conventional methods, such as a high compactness, a small footprint, and superior permeate quality with a high recovery ratio [4]. Considering the highly variable pore size of commercially available UF and nanofiltration membranes, they have great potential to remove particles, contaminants, bacteria, and pathogens, thus improving the overall quality of water [5,6].

The bottlenecks of membrane filtration processes are fouling [7] and concentration polarization [8]. Fouling occurs when transmembrane pressure is increased to maintain a specific permeate flux or when the flux is decreased while the system is maintained at a constant pressure [9]. It can also occur when objects such as suspended solids, germs, and organic compounds accumulate within the membrane pores or on the membrane surface, yielding a declined permeate flux [10]. Fouling is considered irreversible when foulants, which are the materials that cause fouling, adhere to the pores of the membrane. Fouling is considered reversible when foulants adhere only to the surface of the membrane and form a cake layer which hinders the movement of permeate [11]. Unavoidably, membrane fouling always causes a flux decrease, thereby hindering membrane performance and resulting in lower productivity [12,13]. Fouling reduces the active surface, filtration area, and efficiency of membranes, resulting in a shorter membrane lifespan and increased operating and replacement costs [14]. Consequently, characterizing membrane fouling mechanisms and reducing fouling are important research topics [15].

There is great interest in alternative membrane materials, architectures that can decrease fouling, and the necessary chemicals to clean fouled surfaces. The fundamental objective is to decrease the interactions between the pollutants and the membrane surfaces [16]. This can be done either by altering the wetting behavior of membranes [17] or by increasing the fluid turbulence at the membrane surface via surface structuring [18]. The latter approach is favored as it enables the use of commercial membrane materials. Turbulence is predominantly achieved by inducing vortexes in the vicinity of the membrane surfaces and thus utilizing the existence of regular or irregular structure patterns [19]. For membrane-based water and wastewater treatment and desalination, 3D printing offers membranes and customized materials better fouling resistance and productivity [20]. For example, a unique miniature membrane reactor was created and constructed in [21] using novel stereolithography-based 3D printing. This membrane reactor had a smaller volume and a shorter distance between the electrodes compared to computer numerical control-manufactured reactors. Researchers could benefit from performing laboratory measurements in a more economical way by optimizing experimental parameters. Moreover, the membrane filtration performance was strongly affected by the hydrodynamic flow conditions in the feed channel, especially near the membrane surface. These conditions also significantly influenced the mass transport of the bulk liquid toward the membrane surface [22]. The effect of hydrodynamic conditions in the membrane module has been investigated by several researchers [23,24]. The mixing efficiency, flow conditions, and mass and convective transport through the membrane into the module have been improved by appropriately changing these conditions. On the one hand, more intense mechanical stirring inside the module can alleviate membrane fouling by increasing the shear rate on the surface of membranes. On the other hand, simple plastic spacers or three-dimensional-printed (3DP) turbulence promoters can serve as obstacles by disrupting the laminar flow profile in the boundary layer. Intensifying mixing in such a way results in higher speeds and shear rates at membrane surfaces, which can decrease the membrane fouling tendency.

Turbulence promoters can be prepared as two-dimensional structures, but the 3D preparation technique enables significantly more detailed and complex geometries. 3DP can be used to make 3D solid objects from a digital file, while fused deposition modeling (FDM)-based printers enable the preparation of turbulence promoters of different shapes, types, and designs. These FDM-based printers enable more precise design and rapid fabrication [25]. Furthermore, using 3DP turbulence promoters in the module instead of simple plastics can enhance the mixing efficiency and mass transfer through the membrane. This is achieved by reducing the concentration polarization and membrane fouling rate.

Three-dimensional printing seems to be the most promising approach for overcoming the limitations of traditional spacers and turbulence promoter production. This is due to the nature of 3D printing technology, which enables the creation of patterned structures and patterns with great variety and complexity while keeping high precision [26]. Since the advent of 3DP technology in wastewater treatment, it has become more efficient and cheaper to design 3DP elements that can be integrated into membrane filter modules [27,28]. Moreover, it is predicted that 3DP technologies will be used to create a broader range of membrane module components with improved efficiency due to improvements related to speed, materials, and resolution [29]. During the last few years, innovative 3DP feed spacers have shown great potential for different membrane filtration processes, manifesting in enhanced mass transfer, reduced pressure drops, and minimized biofouling. Some studies have utilized 3D printing technologies to improve the performance of feed spacers and prove that they have an important role in reducing the environmental footprint of membrane separation processes [30]. However, the applicability of a simple laboratory-scale dead-end membrane separation system containing a stirred cell with a 3DP turbulence promoter has not been investigated so far.

We designed and fabricated six different 3DP turbulence promoters in this work to mitigate membrane fouling in a classic dead-end membrane separation cell. These turbulence promoters were inserted into the ultrafiltration device, which was tested with commercially available polymer membranes to increase local shear stress on the membrane surface. We analyzed mathematical fouling models, i.e., resistance-in-series models and Hermia models, to investigate the performance of turbulence promoters in the ultrafiltration system and membrane fouling mechanisms. In this work, the impacts of six module-integrated 3DP turbulence promoters were explored. Their ultrafiltration efficiencies were compared, such as permeate fluxes, different resistances, and membrane rejections, based on organic content. Membrane fouling, permeate flux, and the flux decline mechanism were studied using two mathematical models. The resistance-in-series model was used to determine the location of fouling; that is, whether it took place on the membrane surface or inside the pore. Membrane resistance, total resistance, irreversible resistance, and reversible resistance values were calculated using this model. An attempt was made to identify the dominant fouling process by fitting four models to the experimental data.

## 2. Materials and Methods

### 2.1. Model Dairy Wastewater

The dairy wastewater model was freshly prepared by mixing tap water at room temperature, 5 g L^−1^ skimmed milk powder (InstantPack, Berettyóújfalu, Hungary), and 0.5 g L^−1^ anionic surfactant cleaning agent (Chemipur Cl80, Hungaro Chemicals, Nagycserkesz, Hungary) prior to each experiment.

### 2.2. Laboratory Dead-End Ultrafiltration Cell

A laboratory membrane separation device consisting of a UF cell (Millipore, Germany) with an active membrane surface area of 40 cm^2^ and a polyethersulfone membrane with nominal molecular weight limits (NMWL) of 10 or 150 kDa was used. Before use, all pristine membranes were pre-conditioned by immersing them in distilled water for one night. This laboratory instrument was made of a stainless-steel bottom and borosilicate glass walls. The instrument enabled the instantaneous concentration of samples up to 300 mL. All ultrafiltration experiments were carried out at room temperature at a maximum pressure of 4 bar. The initial dairy wastewater feed volume was 100 mL. Permeates (50 mL) were collected until a volume reduction ratio of two was reached.

### 2.3. Design of 3DP PLA Promoters

The FDM technique was used to create the 3DP turbulence promoters using a Creality printer (CR-10S Pro V2 3D; China) with the following parameters: 0.2 mm layer thickness, 100% infill density, cubic infill pattern, 215 °C printing temperature, and 60 °C bed temperature. The designs were created using the Autodesk Fusion 360 design and Cura software (Ultimaker Cure 5.0.0). Six 3DP polylactic acid (PLA)-based turbulence promoters (promoter 1 (Pr. 1), promoter 2 (Pr. 2), promoter 3 (Pr. 3), promoter 4 (Pr. 4), promoter 5 (Pr. 5), and promoter 6 (Pr. 6)) and five stirring speeds (0, 100, 200, 300, and 400 rpm) were used during UF (Figure 1). All turbulence promoters had the same outer diameter of 65 mm, lower inner diameter of 39 mm, and height of 14 mm. These dimensions ensured a perfect fit into the membrane UF cell directly on the membrane surface. Any movement was prevented by attaching an outer O-ring next to the device wall. Only the number, shape, and angles of the panels were changed.

### 2.4. Mathematical Models Used for Modeling

Mathematical models can be used to predict membrane fouling and determine its mechanism. For this reason, resistance-in-series models and Hermia models were investigated.

The total resistance (*R_T_*) of the resistance-in-series model can be calculated by the following equation (Equation (1)) [31]:(1)$$R_{T}=R_{M}+R_{IRR}+R_{REV}$$

Equation (2) can be used to compute the membrane resistance with purified water flux [31]:(2)$$R_{M}=\frac{\Delta p}{J_{v}\cdot \eta_{v}}$$

The irreversible resistance after filtration was calculated from the water flux after flushing the surface of the fouled membrane [31]:(3)$$R_{IRR}=\frac{\Delta p}{J_{v1}\cdot \eta_{v}}-R_{M}$$

The reversible resistance is determined by the flux value that approximates equilibrium [31]:(4)$$R_{REV}=\frac{\Delta p}{J_{ss}\cdot \;\eta_{p}}-R_{M}-R_{IRR}$$

Hermia (1982) created semi-empirical mathematical models to characterize the reduction in permeate flow related to fouling (Equation (5)). They are based on the traditional constant pressure filtering method. Several studies have employed the Hermia models to study membrane fouling. For example, fouling during the treatment of red plum juice, wastewater, and polyethylene glycol filtration was investigated in [32,33,34]. Cake layer formation was the most common method in these works. The intermediate blocking model best described fouling during the treatment of oil-in-water emulsions [35]. For the ultrafiltration of lean coconut at 1.8 bar, [35] found that the standard blocking model best described fouling. However, at temperatures above 60 °C, the intermediate blocking model was the most relevant. The most diverse findings were obtained for a glycerin solution [36]. Distinct fatty acids and pH levels produced different blocking mechanisms. Furthermore, the Hermia model was employed to characterize ultrafiltrate colloidal suspensions [37] and polysaccharide molecules [38,39]. The Hermia models’ mechanism consist of four types (Figure 2), which can be determined by Equation (5) [40]:(5)$$\frac{d^{2}t}{dV^{2}}=K{\left(\frac{dt}{dV}\right)}^{n}$$

Generally, as showed in Table 1, the complete blocking model (n = 2) illustrates that the particle size is bigger than the membrane pore size, so the pores are entirely blocked. The standard blocking model (n = 1.5) states that because the particle size is significantly smaller than the membrane pore diameter, the particles can penetrate the pores and adhere to the pore walls. This way, the particles can potentially block the pores and reduce the pore volume. Furthermore, the intermediate blocking model (n = 1) demonstrates that the particle size in the feed is the same as the pore size in the membrane. This means that particles do not constantly foul membrane pores and some particles may adhere together. Lastly, the cake layer formation model (n = 0) assumes that the particles are bigger than the membrane pores, causing molecules to aggregate on the membrane surface and develop a permeable layer known as cake [32,35].

### 2.5. Organic Rejection Calculations and COD Measurements

Membrane rejections, which determine the quality of filtrates, can be expressed in percentages for a given component, such as chemical oxygen demand (COD) in the remaining concentrate relative to the starting solution. Equation (10) was used to calculate the organic matter rejection of the membrane based on chemical oxygen demand:(10)$$R=\left(1-\frac{c_{P}}{c_{F}}\right)\cdot 100$$

The organic content of the samples was treated with a COD ET 108 digester at 150 °C for 2 h and then measured with a PC CheckIt photometer (Lovibond, Germany). All measurements were carried out three times.

### 2.6. Reynolds Number Calculation

The Reynolds number in UF experiments, without promoter integration, can be calculated using the following equation:(11)$$Re=\frac{D^{2}\cdot \rho \cdot \omega }{\mu }$$

### 2.7. Computational Fluid Dynamics (CFD) in Autodesk

A CAD model (Autodesk CFD, 2023) was utilized for modeling the velocity field of the stirred UF cell module. The model was developed using the membrane surface flow area near the membrane. Only one stirring speed (400 rpm) and one promoter configuration (Pr. 3.) were simulated to simplify the model.

## 3. Results and Discussion

### 3.1. Permeate Flux Results

As shown in Figure 3a and Figure 4a, the best permeate fluxes of 28.88 L m^−2^ h^−1^ and 54.00 L m^−2^ h^−1^ were obtained at 400 rpm for both membranes (10 kDa and 150 kDa, respectively). Integrating turbulence promoters into our systems enhanced the permeate fluxes of the membranes to various degrees (Figure 3b and Figure 4b). The third tested turbulence, Pr. 3., performed the best for both 10 and 150 kDa membranes with permeate fluxes of 46.22 and 78.5 L m^−2^ h^−1^, respectively. In these cases, the flux enhancements were 60.00 and 45.40%, respectively. On the one hand, no substantial disparity was observed in the presence or absence of Pr. 1., which was also the case for the 150 kDa membrane in the presence or absence of Pr. 1. and Pr. 2. In these cases, the shape of the panels was very different compared to the other promoters. Their area was also smaller, and the angle designed was exactly 90°, which did not prove to be favorable over the promoters that have panels with a smaller angle of inclination. On the other hand, the other turbulence promoter designs (Pr. 4., Pr. 5., and Pr. 6.) showed better results compared to the first two designs (especially compared to Pr. 1.) for both membranes except for Pr. 2. (for the 150 kDa membrane). To sum up, Pr. 3. at 400 rpm stirring speed had the best performance with 60.04% and 45.27% for the 10 kDa and 150 kDa membranes, respectively. [41], which used a perforated spacer, reported similar permeate flux enhancements (75% increase). Similarly [42], which compared custom-made spacers with a commercial feed spacer, reported flux enhancements of 15.5% and 38% during the treatment of brackish water by reverse osmosis and UF experiments, respectively. However, the configuration of the ultrafiltration device they used was different from the one we used in this study. Other researchers reported a permeate flux enhancement of 61.7% for 3DP turbulence promoters compared to an empty channel in a contact membrane distillation module [43].

In all ultrafiltration experiments, a relatively low pressure (4 bar) was used, which resulted in relatively low average fluxes. The initial fluxes were about 200 L m^−2^ h^−1^ at 10 kDa and about 800–900 L m^−2^ h^−1^ at 150 kDa. Based on Figure 4, the fluxes obtained for the 150 kDa membranes were selected to evaluate the changes and are shown in Figure 5. Some ultrafiltration experiments for better readability magnified the initial slopes; the figure presents the results of permeate flux before and during the steady state. Due to concentration polarization and the intense contamination of the membrane at the beginning of UF in the first 10 min, the fluxes were suddenly decreased. After that, a slight flux decreasing tendency was observed. It was found that integrating Pr. 3. into the ultrafiltration cell decreased the filtration time the most.

### 3.2. Ultrafiltration Membrane Rejections

The average COD and COD rejections of the UF membranes are shown in Figure 6 and Figure 7. The values gradually increased with increasing stirring speeds for the 10 kDa and 150 kDa membranes up to 8.95% and 5.56%, respectively. The best performance was obtained at 400 rpm stirring speed (Figure 6a and Figure 7a). In general, cell-integrated 3DP turbulence promoters could only slightly increase the rejection values. The highest enhancements were observed using Pr. 4. for the 10 kDa membrane (6.33%) and Pr. 3. for the 150 kDa membrane (5.00%) (Figure 6b and Figure 7b). Since these differences were rather insignificant, the differences in COD rejections were also not considerable, regardless of the type of turbulence promoter used. However, using Pr. 4. and 400 rpm stirring speed, the total COD rejections increased by 15.28% and 11.50% using the 10 kDa and 150 kDa membranes, respectively, compared to the configuration where no mixing was applied. Another research group developed a 3DP photocatalytic feed spacer coated with β-FeOOH nanorods based on a triply periodic minimal surface architecture. After they combined their spacer with a UF membrane, they obtained an overall rejection of 98% for methylene blue [44].

### 3.3. Resistance Results

The resistance-in-series model was used to evaluate the results of the UF experiments with different turbulence promoters (Figure 8 and Figure 9). According to the total resistance values, Pr. 3. with the 10 kDa membrane and Pr. 1. with the 150 kDa membrane resulted in the lowest values, i.e., the best results. The highest values were observed when mixing was not applied. Based on the percentage ratios, the reversible resistances had the highest ratios among the total resistances in all cases. This means that the pollutants can be easily removed from the membrane surfaces. The reversible resistances also had the highest percentage ratio of the total resistance values, ranging between 88.34 and 97.94 and 92.98 and 97.83) for the 10 kDa and 150 kDa membranes, respectively. According to Ng et al., (2021), the feed spacer might help to minimize membrane fouling by inducing turbulence in the feed water flow [26]. They also reported that 3DP feed spacers outperform most non-3DP feed spacers in terms of flow, pressure drops, and even fouling mitigation. Furthermore, 3DP feed spacers are becoming more widespread in real-world applications, since their production does not require as high a resolution as the manufacturing of membrane filters.

Using optical coherence tomography, Park et al. (2021) discovered that the foulant layer generated by using honeycomb-shaped spacers (119.0 m) is substantially smaller than that formed by using regular spacers (175.5 m). Furthermore, hydraulic cleaning studies show that honeycomb-shaped spacers have a more substantial capacity for overcoming fouling resistances caused by the concentration polarization layer and cake layer, resulting in higher permeate production than standard spacer filtration.

### 3.4. Hermia Model Results

Hermia models were initially used to investigate membrane fouling processes during UF. Typical fouling process may be confirmed by fitting the four models to the experimental data. Figure 10 depicts the values measured for the 10 kDa cut-off membrane at various speeds using Equations (6)–(9) and the best model based on the relevant data. The coefficient of determination (R^2^) is most remarkable for n = 0, i.e., the cake layer creation model. Table 2, Table 3, Table 4 and Table 5 demonstrate the R^2^ values for specific measures. The best-fitting model is represented by the highest R^2^ value, highlighted in red. In all cases, regardless of the cut-off value of the membrane, the mixing speed, and the type of turbulence promoters, the cake layer model is the most suitable. This mechanism happens when dissolved molecules are more significant than the pore size of the membrane. As a result, they cannot enter or move through the pores of the membrane, thus accumulating on the membrane surface and forming a layer of cake. After filtering, the cake layer can be readily removed by flushing it with water. This is supported by the results from the analysis of the resistance values, according to which the values of the reversible resistors are the highest (Figure 8 and Figure 9). The reversible resistances are also shown in Table 2, Table 3, Table 4 and Table 5 for the 10 kDa and 150 kDa membranes. Table 2 shows that at n = 0, the R^2^ values for the 10 kDa cut-off membrane range from 0.9923 to 0.7947. As the mixing speed increases, the R^2^ values decrease. This means that mixing eliminates and thins the cake layer formed on the surface of the membrane. The reversible resistances also follow this declining trend well.

### 3.5. Reynolds Number Results

The Reynolds numbers were 2054, 4108, 6162, and 8215 at 100, 200, 300, and 400 rpm stirring speeds, respectively. They indicate that the velocity region was turbulent without integrating a promoter. With promoters, the shear rate on the surface of the membrane can be further increased.

### 3.6. CFD of Ultrafiltration Cell

Figure 11 shows that the flow profile on the surface of the membrane was not uniform. Thus, the velocity distribution changed in the different parts of the UF device near the membrane. The average velocity between the promoter and the membrane was close to the maximum velocity. On the other hand, integrating a promoter into the module resulted in higher velocity values at the surface of the membrane to a significantly greater extent. The higher velocity regions (i.e., the parts highlighted in red) can explain the higher ultrafiltration fluxes compared to the other experiments.

## 4. Conclusions

In this study, six types of three-dimensional turbulence promoters were printed by FDM technology. It was found that integrating turbulence promoters into ultrafiltration cells can result in better membrane separation efficiencies. The best performance was recorded using the third turbulence promoter type (Pr. 3.) with a 400 rpm stirring speed for both 10 kDa and 150 kDa membranes. With this promoter, the average flux values were increased by about 60.04% for the former membrane and by approximately 45.27% for the latter. Distinctions in the COD rejections of ultrafiltration membranes in the presence and absence of turbulence promoters were investigated. Slightly increasing rates were observed for both the 10 kDa and 150 kDa membranes. When Pr. 3. was applied, increases of 12.67% and 10.64% were observed, respectively. For the same promoter, the resistance-in-series model showed that the total resistances decreased by 77.80% using the 10 kDa membrane and by 76.90% using the 150 kDa membrane. The reversible resistance was the most significant, which meant that the cake layer model best described the fouling mechanism in all cases. These results are supported by the ones obtained using the Hermia models. In this mechanism, the contaminants could adhere to the surface of the membrane without fouling the entrance pores, thus forming a cake layer. The cake layer generation enables the fouling particle layer to be readily removed, allowing the membrane to be cleaned and reused more quickly. Finally, when the optimum 3D-printed turbulence promoter was used, the hydrodynamic flow conditions on the surface of the stirred membrane separation cell could be improved. This finding was supported by computational fluid dynamics modeling results.

## Figures and Tables

**Figure 1 membranes-13-00262-f001:**
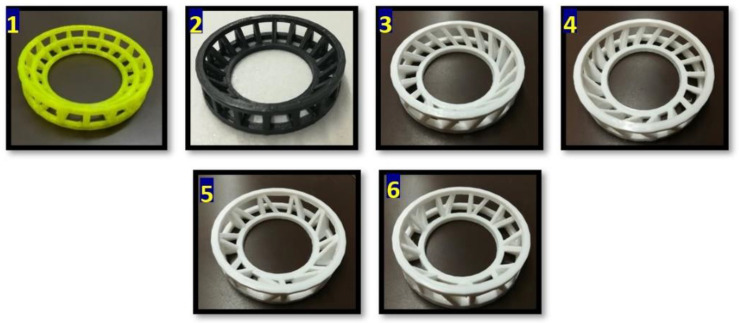
The designed PLA-based 3DP turbulence promoter configurations.

**Figure 2 membranes-13-00262-f002:**
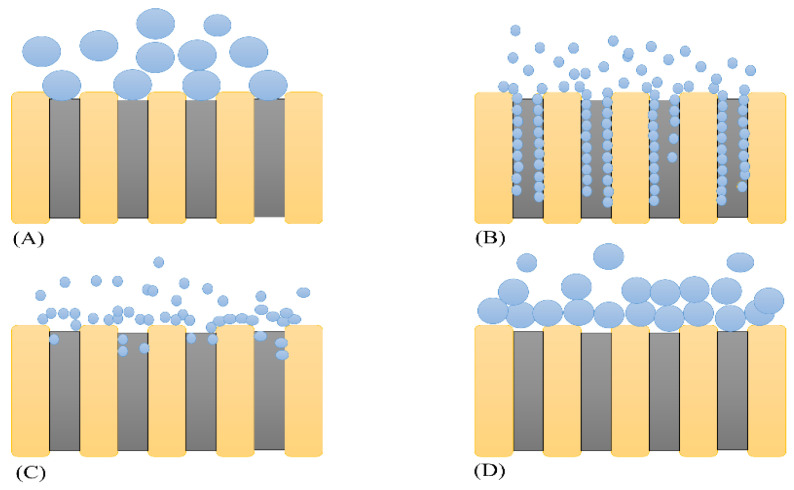
Types of fouling mechanisms: (**A**) complete blocking model, (**B**) standard blocking model, (**C**) intermediate blocking model, and (**D**) cake layer formation model.

**Figure 3 membranes-13-00262-f003:**
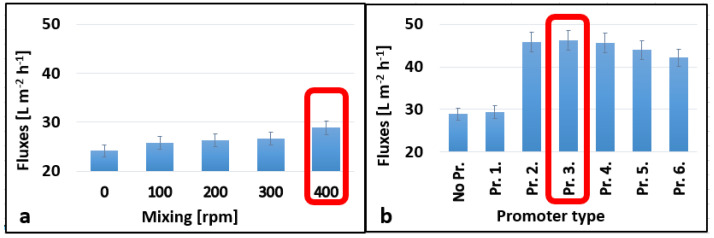
Effect of mixing speeds without 3DP turbulence promoters (**a**) and different promoters at 400 rpm mixing speed (**b**) on the average permeate fluxes for a 10 kDa NMWL UF membrane. A temperature of 25 °C and 4 bar transmembrane pressure (TMP) were applied during the experiments.

**Figure 4 membranes-13-00262-f004:**
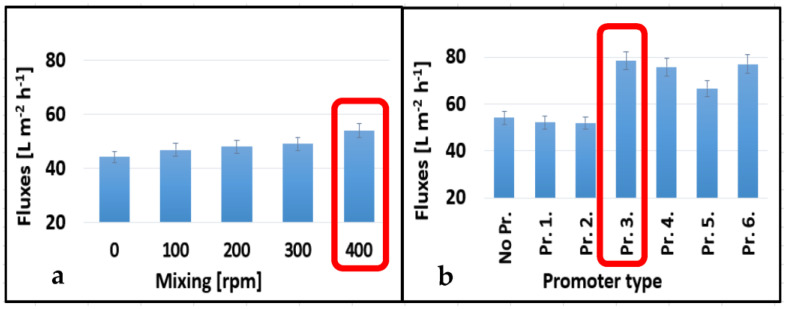
Effect of mixing speeds without 3DP turbulence promoters (**a**) and different promoters at 400 rpm mixing speed (**b**) on the average permeate fluxes of 150 kDa NMWL UF membrane. (T = 25 °C and TMP = 4 bar).

**Figure 5 membranes-13-00262-f005:**
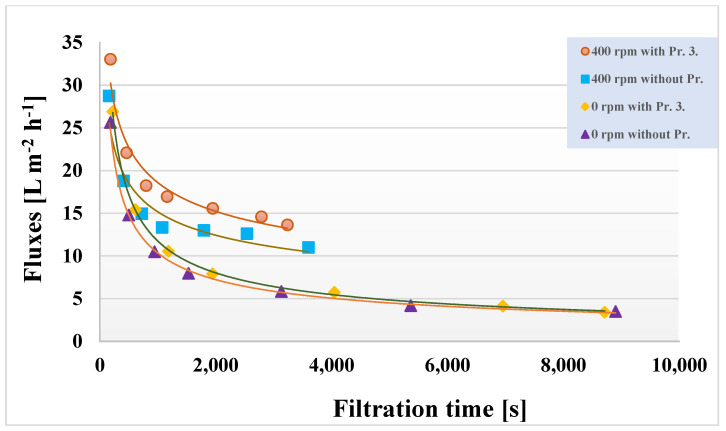
Ultrafiltration permeate flux values of 150 kDa membrane as a function of filtration time (T = 25 °C and TMP = 4 bar).

**Figure 6 membranes-13-00262-f006:**
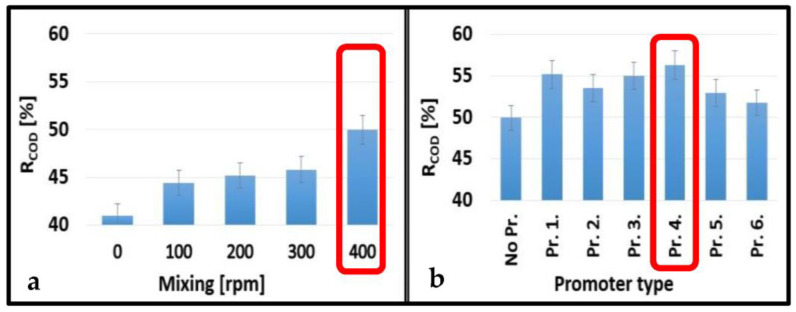
Effects of mixing speeds without 3DP turbulence promoters (**a**) and different promoters at 400 rpm mixing (**b**) on the average COD rejections for a 10 kDa NMWL UF membrane.

**Figure 7 membranes-13-00262-f007:**
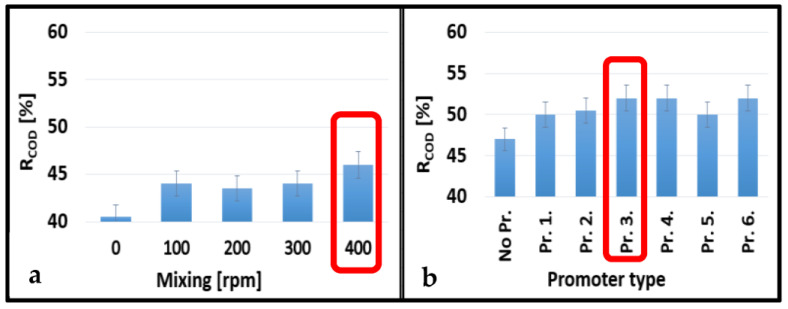
Effects of mixing speeds without 3DP turbulence promoters (**a)** and promoter types at 400 rpm mixing (**b**) on the average COD rejections for a 150 kDa NMWL UF membrane.

**Figure 8 membranes-13-00262-f008:**
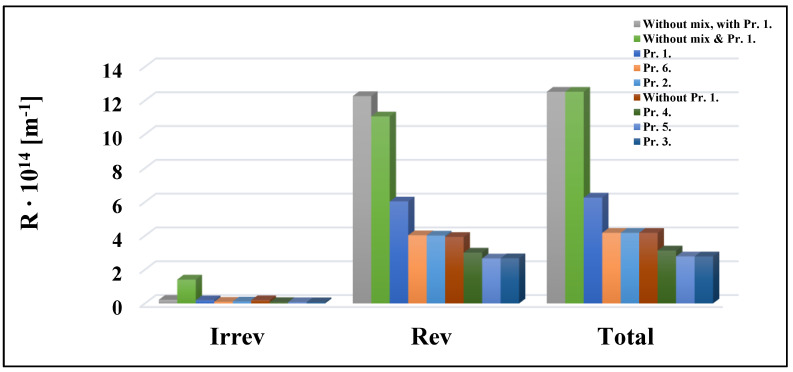
Effects of turbulence promoter types at 400 rpm mixing on the resistance values of a 10 kDa NMWL UF membrane.

**Figure 9 membranes-13-00262-f009:**
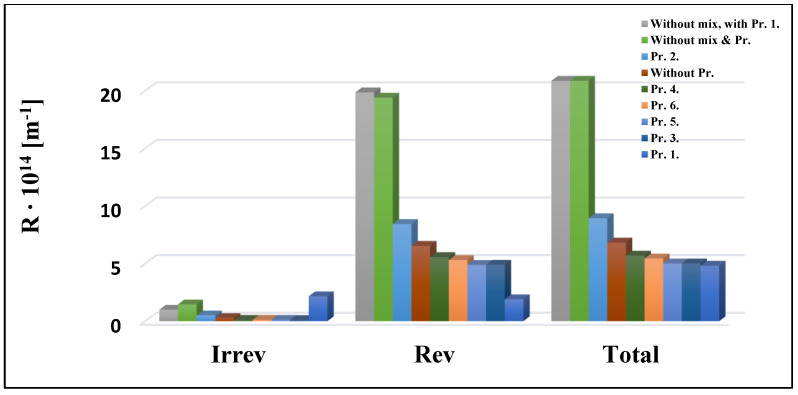
Effects of turbulence promoter types at 400 rpm mixing on the resistance values of a 150 kDa NMWL UF membrane.

**Figure 10 membranes-13-00262-f010:**
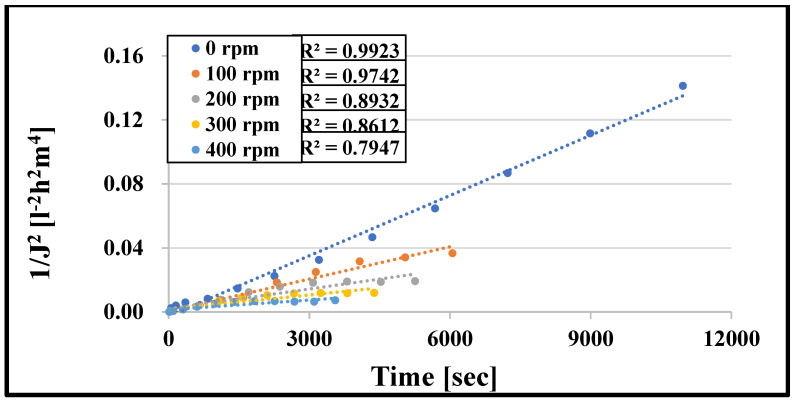
R^2^ values of cake layer formation model (*NMWL* = 10 kDa).

**Figure 11 membranes-13-00262-f011:**
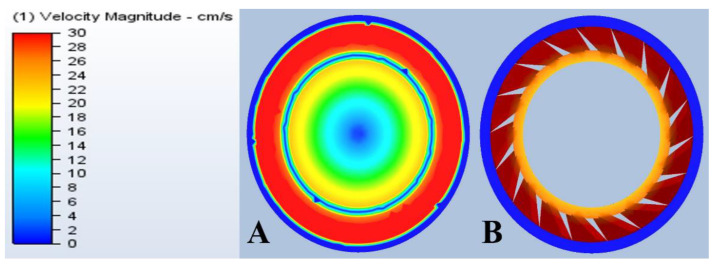
Ultrafiltration cell velocity profiles on (**A**) the surface of the membrane and (**B**) on the surface of the promoter.

**Table 1 membranes-13-00262-t001:** Equations for describing fouling mechanism.

Complete Blocking Model n = 2	Standard Blocking Model n = 1.5	Intermediate Blocking Model n = 1	Cake Layer Formation Model n = 0
$\mathrm{lnJ}={\mathrm{lnJ}}_{0}-k_{c}t$	$\frac{1}{J^{0.5}}=\frac{1}{J_{0}^{0.5}}+k_{s}t$	$\frac{1}{J}=\frac{1}{J_{0}}+k_{i}t$	$\frac{1}{J^{2}}=\frac{1}{J_{0}^{2}}+k_{g}t$
(6)	(7)	(8)	(9)

**Table 2 membranes-13-00262-t002:** R^2^ values of Hermia model and reversible resistance (*NMWL* = 10 kDa).

Stirring Speed (rpm)	n = 2	n = 1.5	n = 1	n = 0	R_rev_
**0 rpm**	0.8618	0.9389	0.9842	**0.9923**	11.06
**100 rpm**	0.6876	0.8242	0.9117	**0.9742**	10.22
**200 rpm**	0.6296	0.7353	0.8124	**0.8932**	8.15
**300 rpm**	0.6201	0.708	0.7772	**0.8612**	6.08
**400 rpm**	0.5859	0.6603	0.7193	**0.7947**	4.23

**Table 3 membranes-13-00262-t003:** R^2^ values of Hermia model and reversible resistance (*NMWL* = 10 kDa).

Turbulence Promoter Type	n = 2	n = 1.5	n = 1	n = 0	R_rev_
**Pr. 1.**	0.8513	0.891	0.9253	**0.9737**	4.9
**Pr. 2.**	0.6206	0.7205	0.8056	**0.9198**	3.44
**Pr. 3.**	0.6509	0.7537	0.8396	**0.9483**	3.46
**Pr. 4.**	0.6367	0.7459	0.8379	**0.9526**	3.92
**Pr. 5.**	0.6337	0.7377	0.8235	**0.9315**	3.45
**Pr. 6.**	0.6395	0.7316	0.8093	**0.9127**	3.43

**Table 4 membranes-13-00262-t004:** R^2^ values of Hermia model and reversible resistance (*NMWL* = 150 kDa).

Stirring Speed (rpm)	n = 2	n = 1.5	n = 1	n = 0	R_rev_
**0 rpm**	0.664	0.8524	0.9601	**0.9945**	14.19
**100 rpm**	0.5788	0.8014	0.9213	**0.9893**	13.52
**200 rpm**	0.5646	0.7695	0.8867	**0.9711**	11.02
**300 rpm**	0.5042	0.6637	0.7701	**0.8791**	7.49
**400 rpm**	0.4846	0.6246	0.7201	**0.8271**	6.54

**Table 5 membranes-13-00262-t005:** R^2^ values of Hermia model and reversible resistance (*NMWL* = 150 kDa).

Turbulence Promoter Type	n = 2	n = 1.5	n = 1	n = 0	R_rev_
**Pr. 1.**	0.546	0.727	0.8509	**0.9584**	6.72
**Pr. 2.**	0.5322	0.6987	0.8125	**0.9312**	4.12
**Pr. 3.**	0.5313	0.6963	0.813	**0.9385**	4.2
**Pr. 4.**	0.5548	0.7036	0.8153	**0.9416**	5.09
**Pr. 5.**	0.5823	0.7091	0.8099	**0.9329**	4.19
**Pr. 6.**	0.5842	0.7161	0.8184	**0.9388**	4.87

## Data Availability

Not applicable.

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
