# Peer review of "Modeling of Organic Fouling in an Ultrafiltration Cell Using Different Three-Dimensional Printed Turbulence Promoters"

_membranes, 2023, doi:10.3390/membranes13030262_

Round 1

Reviewer 1 Report

The manuscript introduced three-dimensional printed turbulence promoters for ultrafiltration performance. The work is interesting but still needs to be improved before it can be further evaluated for possible publication.

1 The flux in Fig 3-6 was far below to the flux values for typical commercial membranes. Why?

2 The manuscript only has the flux value, resistance value and fitting results, the flux curve is missing.

3 What is the difference between the various turbulence enhancers, a hydrodynamics-based analysis is necessary.

4 Multi-cycle experiments are necessary.

Author Response

Dear Reviewer,

Many thanks for your accurate feedback, which certainly has enhanced the quality of the manuscript. We have acted closely on your feedback by:

The new version contains the following changes according to your general comments

  1. Improved the English overall the manuscript
  2. Improved the article overall (from the abstract to the references) which described in the following table:

N

General comments

The answers

1

Does the introduction provide sufficient background and include all relevant references?

Must be improved

Paraphrased some sentences in the Introduction and 10 more new citations were added: (Akter & Park,2023); (Al-Shimmery et al., 2019); (Choi et al., 2016); (Dagar et al., 2023); (Guo et al., 2012); (Maruf et al., 2013); (Padaki et al., 2015); (Szabo-Corbacho et al., 2022); (Todisco et al., 2023); (Zhoa et al., 2022).

2

Are all the cited references relevant to the research?

Must be improved

Because of your suggestion we checked them again one by one and we really feel that all of them are relevant, and we added some more real prevalent as we mention in the previous point.

3

Is the research design appropriate?

Must be improved

A list of abbreviations was added after the keywords to improve the readability of the manuscript.

In addition, the manuscript was redesigned by rewritten the Abstract and included Computational fluid dynamics (CFD) results in it. And paraphrased the Hermia part in the Mathematical models for Modelling and inserted a new figure (our design) for Hermia fouling mechanism types.

4

Are the methods adequately described?

Must be improved

For Materials and Methods, some parts were rewritten and we added Reynolds number calculations and velocity profile pictures by CFD in Autodesk.

5

Are the results clearly presented?

Must be improved

Regarding the Results part, many sentences were paraphrased, and we added the fluxes curve its description; In addition, we changed the figures in the resistance part (Figures 8 and 9) according to your suggestion to make it more understandable; as well as enhancing the quality for the figure of R2 values. Moreover, we explained the results of the Reynolds number and the CFD.

6

Are the conclusions supported by the results?

Must be improved

Finally, we have rewritten the conclusion totally.

According to your specific comments and suggestions

  1. The flux in Fig 3-6 was far below to the flux values for typical commercial membranes. Why?

All the PES polymer membranes were pre-conditioned in the same way by submerging them in distilled water for one night. Since all ultrafiltration studies were conducted at a transmembrane pressure of 4 bar, this can result in relatively low average fluxes. Even the beginning fluxes were substantially higher; the flux curves showed slightly different initial values at 10 kDa, around 200 Lm-2h-1, and 150 kDa, about 800-900 Lm-2h-1. Due to concentration polarization and significant membrane fouling in the first 5 minutes of ultrafiltration experiments, the constant flux was considerably less due to a rapid flux decreasing tendency resulting in lower average fluxes.

  1. The manuscript only has the flux value, resistance value and fitting results, the flux curve is missing.

The flux curve was added and described in more detail.

  1. What is the difference between the various turbulence enhancers, a hydrodynamics-based analysis is necessary.

All of the printed turbulence promoters had the same outer diameter of 65 mm, the lower inner part of 39.00 mm, and a height of 14.00 mm, which can perfectly fit into the membrane ultrafiltration cell directly on the membrane surface to prevent the movement by attach the outer o-ring next to the device wall. Only the number, shape, and angles of the panels were changed; all this information was added to the manuscript. Moreover, for the hydrodynamics-based analysis, a CFD in Autodesk was used to create velocity profile pictures that were added and described.

  1. multi-cycle experiments are necessary.

Some experiments were repeated, after your suggestion, five times, and no change was observed in the flux, rejection, and resistance results. On the other hand, no decomposition of the promoter material was recorded after the experiments, and we did not notice any residues of the promoter PLA material in the concentrated part after the termination of the experiments. Moreover, the promoter dimensions were measured (using Digital Slide Caliper after) in the fifth experiment, and no difference was recorded.

And please find the attached newer version of our manuscript.

Reviewer 2 Report

COMMENTS TO AUTHOR:

The paper entitled “Modeling of organic fouling in an ultrafiltration cell using different three-dimensional printed turbulence promoters”

   I think it is of great interest in the community of membrane. As a result, I will recommend the publication of this manuscript to accept it followed by some minor corrections.

Comments.

1.    Please check English in overall manuscript.

2.    Please improve the Figure 7, 8. Its hard to understand figure.

3.    Please improve the figure 9 quality.

4.    Please make same format of the reference.

Author Response

Dear Reviewer,

Thanks for your accurate feedback, which certainly has enhanced the quality of the manuscript. We have acted closely on your feedback by:

The new version contains the following changes according to your general comments;

Improved the article overall (from the abstract to the references) which described in the following table:

N

General comments

The answers

1

Does the introduction provide sufficient background and include all relevant references?

Can be improved

Paraphrased some sentences in the Introduction and 10 more new citations were added: (Akter & Park,2023); (Al-Shimmery et al., 2019); (Choi et al., 2016); (Dagar et al., 2023); (Guo et al., 2012); (Maruf et al., 2013); (Padaki et al., 2015); (Szabo-Corbacho et al., 2022); (Todisco et al., 2023); (Zhoa et al., 2022).

2

Are all the cited references relevant to the research?

Can be improved

Because of your suggestion we checked them again one by one and we really feel that all of them are relevant, and we added some more real prevalent as we mention in the previous point.

3

Is the research design appropriate?

Can be improved

A list of abbreviations was added after the keywords to improve the readability of the manuscript.

In addition, the manuscript was redesigned by rewritten the Abstract and included Computational fluid dynamics (CFD) results in it. And paraphrased the Hermia part in the Mathematical models for Modelling and inserted a new figure (our design) for Hermia fouling mechanism types.

4

Are the methods adequately described?

Can be improved

For Materials and Methods, some parts were rewritten and we added Reynolds number calculations and velocity profile pictures by CFD in Autodesk.

5

Are the results clearly presented?

Can be improved

Regarding the Results part, many sentences were paraphrased, and we added the fluxes curve its description; In addition, we changed the figures in the resistance part (Figures 8 and 9) according to your suggestion to make it more understandable; as well as enhancing the quality for the figure of R2 values. Moreover, we explained the results of the Reynolds number and the CFD.

6

Are the conclusions supported by the results?

Can be improved

Finally, we have rewritten the conclusion totally.

According to your specific comments and suggestions

  1. Please check the English in overall manuscript.

After we had rewritten the overall manuscript based on the reviewers' suggestions, an official English translator improved the manuscript's English language.

  1. Please improve Figures 7 and 8. It is hard to understand the figure.

Figures 7 and 8 were changed, improved, and made more understandable, as well as explained in the text in more detail. 

  1. Please improve the figure 9 quality.

Because of your suggestion, we improved the quality of figure 9.

  1. Please make the same format of the reference.

We checked them one by one and unified them by Mendeley, so now we think this part is in the same format as you suggested.

And please find the attached newer version of our manuscript.

Round 2

Reviewer 1 Report

no further comments